# Impact of Repetitive Transcranial Magnetic Stimulation (rTMS) on Theory of Mind and Executive Function in Major Depressive Disorder and Its Correlation with Brain-Derived Neurotrophic Factor (BDNF): A Randomized, Double-Blind, Sham-Controlled Trial

**DOI:** 10.3390/brainsci11060765

**Published:** 2021-06-09

**Authors:** Jie Tong, Jie Zhang, Ying Jin, Weiqing Liu, Hao Wang, Ying Huang, Dianhong Shi, Minhuan Zhu, Na Zhu, Tingting Zhang, Xirong Sun

**Affiliations:** Shanghai Pudong New Area Mental Health Center, Tongji University School of Medicine, Shanghai 200124, China; tongj@shspdjw.com (J.T.); zhangj@shspdjw.com (J.Z.); Jinying@shspdjw.com (Y.J.); liuwq@shspdjw.com (W.L.); Wangh@shspdjw.com (H.W.); huangy@shspdjw.com (Y.H.); Shidh@shspdjw.com (D.S.); Zhuminhuan@shspdjw.com (M.Z.); Zhun@shspdjw.com (N.Z.); zhangtt@shspdjw.com (T.Z.)

**Keywords:** major depressive disorder, repetitive transcranial magnetic stimulation, theory of mind, executive function, brain-derived neurotrophic factor

## Abstract

Background: Studies have implicated hypofrontality in the pathogenesis of impaired theory of mind (ToM) and executive function (EF) in major depressive disorder (MDD). These symptoms are usually resistant to treatment. Repetitive transcranial magnetic stimulation (rTMS) has been shown to reverse hypofrontality. Moreover, BDNF is an effective biomarker of antidepressant effects, but there have been very few studies on the correlation between BDNF and rTMS. We aimed to evaluate the efficacy of 20 sessions of a 10 Hz unilateral rTMS intervention over the left dorsolateral prefrontal cortex (DLPFC) in improving ToM and EF in patients with MDD and its correlation with BDNF. Methods: A total of 120 MDD patients were enrolled in this randomized, sham-controlled, double-blind trial. Each participant received 20 sessions of rTMS at 10 Hz frequency through the active or the sham coil over 4 weeks. ToM was assessed with the facial emotion identification test (FEIT) and hinting task (HT). EF was assessed with the Wisconsin card sorting test (WCST). BDNF assessments were carried out at baseline and 2-, 4-, 12-, and 24-week follow-ups. Results: The improvement in the ToM (FEIT, HT) in the active rTMS group was significantly different from that in the sham rTMS group (F = 18.09, *p* < 0.001; F = 5.02, *p* = 0.026). There were significant differences in the WCST (categories completed, response errors, response perseverative errors, non-response perseverative errors) after logarithmic transformation at different time points in the active rTMS group (F = 14.71, *p* < 0.001; F = 5.99, *p* = 0.046; F = 8.90, *p* = 0.031; F = 2.31, *p* = 0.048). However, there was no significant difference in log transformed BDNF concentration between the two groups (*t* = 0.07 to *t* = 1.29, *p* > 0.05). BDNF was negatively correlated with WCST categories completed at the 24th week (*r* = −0.258, *p* = 0.046). Conclusions: The results show that rTMS may improve the ToM and EF of patients with MDD and there was no significant correlation with serum BDNF concentration. RTMS can not only be used for treatment of patients with MDD but also has a positive effect on ToM and EF.

## 1. Introduction

Major depressive disorder (MDD) is a recurrent severe affective disorder with high prevalence, high recurrence rate, and high suicide and disability rates [1]. According to the 2017 global burden of disease survey, MDD has become one of the three major causes of disability in all age groups worldwide [2]. Years lost due to disability (YLDS) caused by MDD accounted for 14.3% of the total YLDS [3]. The World Health Organization (WHO) survey shows there are approximately 350 million patients with depression worldwide, and the lifetime prevalence rate is 10–20% [4].

MDD often results in negative interpersonal experience which, in turn, usually causes depressed subjects to isolate themselves, perpetuating their depression, and which can precede and increase depressive symptomatology [5,6]. These symptoms have been demonstrated to involve the limbic system and prefrontal cortex (PFC), especially the amygdala and the hippocampus [7]. Changes in the activity of these brain areas are considered to be key to the pathophysiology of depression [8]. Therefore, abnormal activation of these brain regions has been related to the severity of depressive symptoms, such as the relationship between depression and learning, memory and emotional dependence [9].

There are significant theory of mind (ToM) and executive function (EF) deficits in MDD [10,11]. Broadly construed, ToM refers to the daily ability of an individual to infer or decode others’ psychological states to adapt one’s own behavior, and to understand or predict other people’s behavior, such as beliefs, emotions, and intentions [12]. It is considered to be a highly specialized and human-specific skill that constitutes a core aspect of social cognition and an important prerequisite to function in social groups [13]. EF is a term consisting of a series of cognitive processes, including the ability to formulate plans and goals and perform effectively [14]. Studies that have begun to correlate depressive symptoms and deficits in EF involve neuroimaging and psychometrics. EF deficits have been found in problem solving ability, abstraction ability, loss of cognitive flexibility, and tendency to perseverate [15]. ToM and EF are often associated with deficits in frontal regions and depressive symptoms are characterized by frontal dysfunctions [16,17].

Repetitive transcranial magnetic stimulation (rTMS) is a noninvasive brain stimulation technique that has been proven to be able to treat a variety of psychiatric and neuropsychiatric disorders, including depression, PTSD, and migraine [18]. In 2008, the US Food and Drug Administration (FDA) cleared the NeuroStar TMS Therapy System as the first device for rTMS treatment of MDD, and it has been adopted into clinical practice [19,20]. The acute antidepressant properties of rTMS delivered to the dorsolateral prefrontal cortex (DLPFC) have been extensively examined [21]. rTMS can significantly modulate dopamine and serotonin content and turnover rates in the brain, modulate cortical beta-adrenergic receptors, and decrease 5-HT2 receptors in frontal cortex. The National Institute of Mental Health (NIMH) showed that the acute antidepressant properties of rTMS targeting the DLPFC were effective [22].

Brain-derived neurotrophic factor (BDNF) is a small molecule dimer protein that can promote the growth and differentiation of neurons [23]. Studies have shown that BDNF is associated with depression in animal models [24]. Serum BDNF was increased after administration of antidepressants, which could be reversed by central administration of BDNF [25,26]. Duman [27] et al. formulated the “neurotrophic hypothesis of depression” and believed that depression was caused by a lack of BDNF in the brain. BDNF has become an important biomarker in the pathogenesis and treatment of depression [28].

The literature on the effects of rTMS on ToM and EF of MDD consists mainly of studies with very small sample sizes and inadequate power to detect intergroup differences [29,30]. Moreover, although the role of BDNF in the pathogenesis of MDD is relatively clear, its correlation with ToM and EF has not been confirmed. We aimed to evaluate the efficacy of 20 sessions of a 10 Hz unilateral rTMS intervention over the DLPFC in improving ToM and EF in patients with MDD. ToM and EF were assessed in a randomized, double-blind trial by a facial emotion identification test (FEIT), hinting task (HT), and Wisconsin card sorting test (WCST) and compared to sham controls.

## 2. Methods

### 2.1. Study Design

The study was conceptualized as a randomized, sham-controlled, double-blind trial. The study was conducted at the Shanghai Pudong New Area Mental Health Center (PMHC), Tongji University School of Medicine, which has a dedicated rTMS unit with a fully functioning rTMS lab since 2006. We have been running this rTMS clinic everyday. The sample size of the program was calculated according to the formula $N_{sample}={\left[\frac{Z_{a/2}\sqrt{\pi_{0}(1-\pi )}+Z_{\beta }\sqrt{\pi (1-\pi )}}{\pi -\pi_{0}}\right]}^{2}$ [31]. The significance level was 0.05, and the expected effective rate was 80%. The estimated sample size was 48 per group. Our study included a sample of 60 per group, accounting for approximately 20% losses in follow-up (those not discharged automatically according to the discharge standards or that could not be completed for other special reasons).

### 2.2. Randomization Procedure

For randomization of enrolled patients, 120 sequentially numbered opaque envelopes with either ‘sham’ or ‘active’ written on it were prepared using a computer generated random sequence by the statistics department of PMHC. Only two rTMS technicians had access to them. As per the random sequence, they would allot a number to the randomized patient and provide the intervention as mentioned on the card inside. The subjects were blinded regarding the treatment being given, and the rater and lab technician were blinded to the allocation. Only the rTMS trained technician who administered rTMS was aware of the allocation from time of randomization to time of data analysis [32].

### 2.3. Participants

All participants were inpatients in the Department of Mood Disorders of PMHC, Tongji University School of Medicine, from January 2018 to December 2019. Inclusion criteria: (1) patients meeting the DSM-5 [33] diagnostic criteria for major depressive disorder; (2) age 18 to 70 years, inclusive; (3) Hamilton Depression Scale (HAND-17) score > 24; (4) only escitalopram was used at a dose of 20 mg per day for more than one month; (5) having a certain visual and auditory resolution without understanding disorders; (6) ability to complete the test independently; (7) more than primary school education; (8) both patients and guardians agreed to participate in the study and signed the informed consent. Exclusion criteria: (1) history of severe physical disease or brain organic disease; (2) history of brain injury or head surgery; (3) history of metal implantation in the body; (4) failure to receive treatment with MECT or rTMS within one month; (5) history of epilepsy and drug dependence; (6) use of other antidepressants or with drug dose changes during the observation period; (7) combined with other mental disorders.

### 2.4. Repetitive Transcranial Magnetic Stimulation (rTMS)

We used three RT-100 repetitive transcranial magnetic stimulation therapeutic instruments produced by Sichuan Junjian Wanfeng Medical Instrument Company. Before the intervention, the data and equipment of all rTMS therapeutic instruments were calibrated by the manufacturer. For the active group, rTMS Magstim’s 70 mm figure-of-8 Double Rapid2 Air Cooled Coil (P/N 3910-00 S/NO 728) was used, while for sham group, Magstim’s sham coil (P/N 3950-00 S/NO 105) was used. The coils were identical in external appearance which ensured that the patients remained blind to the intervention. Stimulation frequency: 10 Hz; intensity: 120% resting motor threshold (RMT); intertrain time interval: 90 s to 120 s (as calculated automatically by the TMS machine for built safety measures); total of 2000 pulses imparted in 10 trains; location: left DLPFC. Both sham and active group patients received the same stimulation parameters. We defined the site of stimulation as a point 5.5 cm anterior and then 0.5 cm lateral from the point where optimum stimulation of the right abductor pollicis brevis muscle was observed on stimulation of cortical representation of the left cerebral cortex. A number of studies show that the DLPFC is the most important emotion regulation area and the best choice for rTMS treatment; the treatment time was 20 min and treatment position was either sitting or semi-reclining. Before treatment, patients must wear noise reduction earplugs and remove all objects sensitive to magnetic field (such as mobile phones, wrist watches, jewelry, credit cards, hearing aids, metal denture, etc.). The patients were treated once a day, 5 times per week, for 4 weeks, yielding a total of 20 times. The same procedure was followed for both groups of patients. The rTMS technician had a bachelor’s degree in rehabilitation medicine or a master’s degree in psychiatry and was a licensed Chinese rTMS therapist.

## 3. Measures

### 3.1. 17-Items Hamilton Depression Rating Scale (HAMD-17)

The HAMD-17 scale was proposed by Hamilton in 1960 [34]. It is the most commonly used scale to evaluate depression. The HAMD-17 scale includes 17 items. According to standardized description, these items have to be scored on three- or five-point scales. The measurement indexes include:(1) depressed mood; (2) guilt; (3) suicide; (4) insomnia, initial; (5) insomnia, middle; (6) insomnia, early morning; (7) work and interests; (8) retardation; (9) agitation; (10) anxiety, psychic; (11) anxiety, somatic; (12) gastrointestinal symptoms; (13) general somatic symptoms; (14) loss of libido; (15) hypochondriasis; (16) weight loss; (17) loss of insight. Mild depression: the total score was 8–17; moderate depression: the total score was 18–24; severe depression: the total score >24 [35].

### 3.2. Facial Emotion Identification Test (FEIT)

The FEIT consists of 19 black-and-white photos of the faces of different individuals, each of which depicts one of six different emotions (happiness, sadness, anger, surprise, fear, shame), shown one at a time for 15 s, with 10 s of blank screen between each stimulus presentation. The 15 photos depict negative emotions (sadness, anger, fear, shame), while 4 photos depict positive emotions (happiness, surprise). In the management of FEIT for this study, each participant was presented with these photos of facial emotions on a laptop screen. The participants were provided with an answer form with 19 items, each with six emotional choices. After each stimulus, the participants were required to choose which of the six emotions was depicted in the picture and to mark it in the form. The total test score was calculated as the number of correct answers (0–19). The score for positive emotions was calculated as the number of correct answers for positive emotions (0–4), and the score for negative emotions was calculated as the number of correct answers for negative emotions (0–15) [36].

### 3.3. Hinting Task (HT)

HT is a linguistic measure of ToM which evaluates the participants’ ability to infer the real intentions behind indirect speech utterances or hints. The task consisted of 10 essays that involved dropping a hint; participants were asked to say the intention of the role dropping the hint [37]. Repeated reading was allowed if required by the participant. If the participant’s first attempt was correct, a score of two was given. If the participant gave the correct response when prompted with paraphrasing of the hint, a score of one was given. The failure score to infer the true intention was scored as zero. Scores were added to obtain the total score, which ranged from 0 to 20, where higher scores indicate better ToM performance (exploring the use of the hinting task in first episode psychosis). Box 1 shows one of the 10 items on the HT [38].

### 3.4. Wisconsin Card Sorting Test (WCST)

The WCST is a neuropsychological test that is often used to evaluate the executive function of the frontal lobe in patients with mental disorders and brain organic diseases. The ability of classification, concept formation, learning and memory, and cognitive transfer are measured. The WCST was proposed by Berg in 1948 and revised by Heaton [39]. The WCST consists of two card packages, each of which has four stimulus cards and 128 response cards. Each card has various geometric shapes (triangle, five pointed stars, cross, circle) in different colors (red, green, yellow, blue) and numbers (1, 2, 3, 4). The participants were asked to accurately sort each response card with one of four stimulus cards based on a rule when given feedback (right or wrong). The measurement indexes of cognitive function include the following. (1) Categories completed (CC): number of categories completed after measurement. The score range was 0–6, which was used to measure the degree of mastering the concepts classified into different categories; (2) Response errors (Re): all responses that do not meet the required response principles. A normal value ≤ 45 indicates the cognitive transfer ability of the subjects; (3) Response perseverative errors (RPE): participants cannot give up the original classification principle. After a change of classification principle, they stubbornly continue to classify according to the original classification principle. It can reflect the formation of concepts, the use of correction and the plasticity of concepts; (4) Nonperseverative response errors (nRPE): the difference between the total number of errors and the number of persistent errors. A normal value ≤ 24 and a high score indicates inattention or confusion [40]. The test was carried out by the stand-alone version of the WCST software system developed by Shanghai Ruanlong Technology Development Co. Ltd.

### 3.5. Procedure

All the enrolled participants’ legal guardians/next of kin provided written informed consent. After baseline assessment on the mentioned instruments, rTMS intervention was provided as per the mentioned protocol. The same assessments and tests were also carried out at 2, 4, 12, and 24 weeks. Subsequently, an experimenter instructed the participants to complete the test in a quiet room by themselves through computer. The experimenter checked the completion of each item. All evaluators were trained for consistency. In addition, the participants provided 4 mL of whole blood from the elbow vein at 6:00 the next morning for BDNF detection. The samples were stored at −70 °C and sent to the BDNF Laboratory of PMHC. The serum concentration of BDNF was determined by an enzyme-linked immunosorbent assay.

### 3.6. Data Analysis

Data were analyzed using SPSS version 22.0 (SPSS, Inc., Chicago, IL, USA) statistical software. Descriptive analysis was performed for sociodemographic data. Measurement data was expressed as mean ± standard deviation (*χ* ± sd). For the comparison between groups of measurement data with a normal distribution, T-test and one-way and two-way repeated measure analysis of variance (ANOVA) were used. *χ*^2^ test was used for technical data. Spearman correlation coefficient was used. *p* < 0.05 as the difference was considered statistically significant.

### 3.7. Ethics

The study was approved by the Ethics Committee of the Shanghai Pudong New Area Mental Health Center and Tongji University Mental Health Center (No.: 201711). All procedures were carried out in accordance with the ethical standards of the Human Experimentation Responsibility Committee (institutional and national) and the Helsinki Declaration of 1975, as revised in 2008. The trial protocol was registered with the Chinese Clinical Trial Registry (CCTR) with the number ChiCTR2100044633.

## 4. Results

A total of 347 patients were screened for inclusion; 120 were randomized to the active rTMS group (*n* = 60) or sham rTMS group (*n* = 60). An attrition rate of 3.3% (*n* = 2) in the active rTMS group and 5% (*n* = 3) in the sham rTMS group was seen during the intervention. This further increased to 8.3% in active rTMS group and 11.6% in sham rTMS group. The number of dropouts in the two groups was statistically similar (*p* > 0.05) (Figure 1).

### 4.1. Demographic Characteristics

Table 1 shows the baseline demographic characteristics of the participants. No significant differences were found between the active and sham rTMS groups in demographic characteristics at the baseline (*p* > 0.05).

### 4.2. Safety and Tolerability

The adverse events of rTMS were evaluated by case self-report. One patient in the active rTMS group could not tolerate the noise of the instrument during the 3rd session, after which the intervention was discontinued. Other than this event, both active and sham interventions were well tolerated.

### 4.3. Analysis of Depressive Symptoms

HAMD-17 values of the two groups were compared by one-way repeated measures analysis of variance. The results showed that the HAMD-17 scores of the active rTMS group and sham rTMS group decreased after the intervention, and the difference between the two groups was statistically significant (F = 183.88, *p* < 0.001; F = 68.61, *p* < 0.001). HAMD-17 values of the two groups at different time points were compared. The results show that the score of active rTMS group was significantly lower than that of sham rTMS group, and the difference was statistically significant at 4, 12, and 24 weeks (*p* < 0.001). HAMD-17 values of different groups at different time points were compared by two-factor repeated measures analysis of variance. The results showed that there were significant differences in the main effects at time points, groups and time and groups (*p* < 0.001) (Table 2).

### 4.4. Analysis of ToM

FEIT and HT values of the two groups at different time points were compared by one-way repeated measures analysis of variance. There were significant differences in FEIT and HT after logarithmic transformation at different time points in the active rTMS group (F = 18.09, *p* < 0.001; F = 5.02, *p* = 0.026). However, there were no significant differences in FEIT and HT in the sham rTMS group (F = 2.32, *p* = 0.093; F = 1.37, *p* = 0.141). By comparing FEIT and HT between the two groups at different time points, the results showed that the mean of the two groups increased, but the active rTMS intervention was more effective than the sham rTMS at different time points (Table 3).

### 4.5. Analysis of EF

WCST values of two groups at different time points were compared by one-way repeated measures analysis of variance. As Table 4 shows, there were significant differences in the four subcategories of WCST after logarithmic transformation at different time points in the active rTMS group, including the CC, RE, RPE, and nRPE. The RE, RPE, and nRPE in the AG were significantly different (F = 5.99, *p* = 0.046; F = 8.90, *p* = 0.031; F = 2.31, *p* = 0.048) and the CC in the AG was significantly different (F = 14.71, *p* < 0.001). However, there was no significant difference for the four subcategories in the sham rTMS group (F = 1.14 to F = 2.52, *p* > 0.05). By comparing four subcategories of WCST between the two groups at different time points, the results showed that the mean of the two groups increased, but the active rTMS intervention was more effective than the sham rTMS group at different time points.

### 4.6. Analysis of BDNF

Because BDNF values had a skewed distribution, logarithmic transformation was first carried out to make them normally distributed. The log transformed BDNF values of the two groups at different time points were compared by one-factor repeated measures analysis of variance. The results show that the log transformed BDNF concentration of the two groups increased after intervention, and the differences were significant (F = 31.84, *p* < 0.001; F = 17.65, *p* < 0.001). The log transformed BDNF concentration was compared between two groups at different time points. The results show that there was no significant difference in log transformed BDNF concentration between the two groups at different stages of treatment (*t* = 0.07 to *t* = 1.29, *p* > 0.05). The log transformed BDNF values of different groups at different time points were compared by two-factor repeated measures analysis of variance. The results show that there was a main effect at time (F = 10.70, *p* < 0.001), but there were no significant differences between the groups nor an interaction between time and groups (F = 0.52, *p* = 0.471; F = 1.55, *p* = 0.350) (Table 5).

### 4.7. Correlation Analysis of BDNF with Depressive Symptoms, ToM, and EF

The correlation between BDNF concentration and HAMD-17, FEIT, HT, and WCST score in the active rTMS group was analyzed at each time point before and after intervention. The results showed that BDNF was negatively correlated with HAMD-17 and CC of WCST at the 24th week (*r* = −0.788, *p* = 0.035; *r* = −0.258, *p* = 0.046). There was no significant correlation at other time points (*p* > 0.05) (Table 6).

## 5. Discussion

We examined the impact of 20 courses of 10 Hz rTMS over the left DLPFC in the treatment of ToM and EF in MDD and the correlation with BDNF in a randomized, double-blind, sham-controlled trial. We further confirmed the therapeutic effect of rTMS on depressive symptoms in MDD patients and found that rTMS also had good therapeutic effect on ToM and EF in MDD patients. Significant differences were seen in ToM and EF, as measured by FEIT, HT, and WCST, between the active and sham groups after the rTMS intervention. Our study has one of the largest sample sizes of published studies evaluating rTMS interventions for ToM and EF of MDD available to date and, hence, is adequately powered to detect small differences. Previous studies have used rather small samples and nonrandomized controlled trials; a maximum of 45 participants were included in the 1 Hz rTMS intervention [41]. Furthermore, we found only three RCTs (published in English language) that assessed the effect of rTMS on ToM and EF [42,43,44]. There are suggestions from the literature that rTMS interventions with frequencies less than 10 Hz might lead to better improvement in ToM and EF [45,46,47,48]. Our results provide further validation of these findings. The importance of the results of our study is reiterated by the fact that we followed our patients for a substantially long duration, and all of our participants were hospitalized patients. Therefore, the attrition rate in our study was low.

The neurotrophic factor hypothesis of depressive disorder suggests that antidepressant therapy plays a role by increasing the expression of BDNF in the brain, enhancing prominent plasticity and promoting the survival of neurons [27]. Björkholm [49] et al. found that BDNF was lower in patients with depression than in the control group. After antidepressant treatment, BDNF returned to the normal level. There are few randomized controlled studies on the impact of rTMS on BDNF in patients with depression. Heath [50] et al. treated murine models of depression with low-, medium-, and high-frequency rTMS. The results showed that the level of BDNF in frontal cortex and hippocampus of medium-frequency rTMS group was significantly higher than that of the control group. Gedge [51] et al. conducted a single-blind study of 29 patients with MDD who received rTMS treatment. This study suggests that there was no significant change in serum BDNF concentration pre- and posttreatment, and rTMS may not play a clinical role by changing serum BDNF levels in patients with MDD. However, the researchers pointed out that the small sample size in this study may affect the conclusion. Our study found that serum BDNF levels of the active and sham rTMS group were significantly higher after the treatment, but there was no significant difference in serum BDNF levels between the two groups at different time points, which may be related to the simultaneous use of citalopram in the two groups. In other words, the synergistic impact of rTMS on antidepressant treatment may not be related to the change in serum BDNF concentration. This is consistent with Brunoni’s meta-analysis and suggests that the BDNF is not an effective biomarker of the antidepressant effect of nonmapping brain stimulation [52].

There was no significant correlation between serum BDNF concentration and FEIT, EF, and WCST at baseline. However, because there was a negative correlation between the CC of WCST at the 24th week, it cannot be excluded that the correlation will gradually become significant with the extension of follow-up time. Xiu MH [53] et al. performed a comparative study of serum BDNF and WCST in 112 long-term hospitalized patients with chronic schizophrenia. Their results suggest that BDNF levels were correlated with EF measured by verbal fluency tests (VFT) and WCST and that BDNF was involved in the pathophysiology of EF impairments in patients with chronic schizophrenia. Lu RB [54] et al. enrolled 170 heroin addicts and 141 healthy controls to measure plasma BDNF and WCST. The study found that that there was a negative correlation between the number of trials completed in WCST and BDNF levels.

Our study has several limitations. First, we did not use rTMS at different frequencies and locations. Moreover, we did not expand more measurement tools for ToM and EF. Future clinical studies should further explore the impact of rTMS with different frequencies and locations on MDD and the correlation between BDNF gene polymorphism [55], methylated BDNF gene [56], the BDNF-Tkrb signaling system and rTMS [57].

## 6. Conclusions

Our study shows the benefits of 10 Hz rTMS over left DLPFC for 20 sessions over 4 weeks for patients suffering from MDD with predominant ToM and EF, and there was no significant correlation with serum BDNF concentration. This study adds to the existing literature due to its robust design and large sample size despite limitations. RTMS can be used as the treatment of patients with MDD, and it is also useful because of its a simultaneous effect on ToM and EF.

## Figures and Tables

**Figure 1 brainsci-11-00765-f001:**
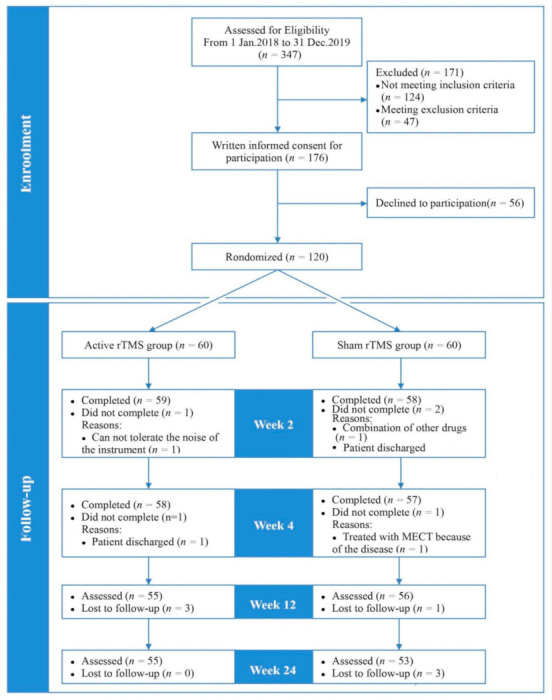
Sample flow diagram depicting the flow of participants through the study.

**Table 1 brainsci-11-00765-t001:** Baseline sociodemographic characteristics of the sample population.

Variable	Active rTMS Group (*n* = 60)	Sham rTMS Group (*n* = 60)	*t*/*χ*^2^	*p*
Age in years (mean ± SD)	48.65 ± 14.04	47.75 ± 13.43	0.361	0.721
Sex, *n* (%)			0.845	0.358
Male	24 (45.28%)	29 (54.72%)		
Female	36 (53.73%)	31 (46.27%)		
Marital status, *n* (%)			1.913	0.137
Unmarried	17 (56.67%)	13 (43.33%)		
Married	28 (45.16%)	34 (54.84%)		
Divorce or widowhood	15 (53.57%)	13 (46.43%)		
Education, *n* (%)			1.048	0.220
Junior high school and below	14 (45.16%)	17 (54.84%)		
High school	19 (55.88%)	15 (44.12%)		
Junior college or above	27 (49.09%)	28 (50.91%)		

**Table 2 brainsci-11-00765-t002:** Comparison of HAMD-17 scores between the two groups across the assessment time points.

Time Points (Week)	AG	SG	*t*	*p*	Time	Group Experimental Group	Time × Group
0	29.63 ± 3.55	28.50 ± 3.97	1.65	0.102	F = 234.27	F = 21.49	F = 1.55
2	24.17 ± 3.16	24.47 ± 3.24	0.51	0.609	*p* < 0.001 *	*p* < 0.001 *	*p* < 0.001 *
4	12.68 ± 2.78	14.55 ± 2.44	3.91	<0.001 *			
12	7.53 ± 3.43	12.78 ± 2.04	10.18	<0.001 *			
24	6.92 ± 2.63	10.67 ± 1.90	8.96	<0.001 *			
F	183.88	68.61					
*p*	<0.001 *	<0.001 *					

* *p* < 0.001.

**Table 3 brainsci-11-00765-t003:** Comparison of FEIT and HT scores between the two groups across the assessment time points.

Time Points (Week)	Facial Emotion Identification Test (FEIT)		Hinting Task (HT)	
	AG	SG	AG	SG
0	19.58 ± 2.87	19.57 ± 2.77	12.46 ± 3.25	12.50 ± 2.45
2	20.28 ± 2.61	19.82 ± 2.51	12.50 ± 3.25	13.13 ± 2.55
4	21.92 ± 2.49	20.20 ± 2.16	13.40 ± 2.12	13.23 ± 2.37
12	22.35 ± 2.43	20.38 ± 1.96	14.35 ± 1.74	13.37 ± 2.26
24	23.12 ± 2.21	20.52 ± 1.83	14.79 ± 1.74	13.62 ± 2.06
F	18.09	2.32	5.02	1.37
*p*	<0.001 **	0.093	0.026 *	0.141

Active rTMS group—AG; Sham rTMS group—SG. * *p* < 0.05. ** *p* < 0.001.

**Table 4 brainsci-11-00765-t004:** Comparison of WCST scores between the two groups across the assessment time points.

Time Points (Week)	Categories Completed (CC)		Response Errors (RE)		Response Perseverative Errors (RPE)		Non-Response Perseverative Errors (nRPE)	
	AG	SG	AG	SG	AG	SG	AG	SG
0	1.97 ± 1.39	1.78 ± 1.39	73.40 ± 21.31	70.53 ± 21.75	36.20 ± 16.19	35.55 ± 18.53	45.60 ± 16.49	46.73 ± 21.14
2	2.35 ± 1.16	1.92 ± 1.38	67.52 ± 16.44	68.73 ± 19.48	33.78 ± 14.26	34.80 ± 17.88	43.82 ± 11.90	46.43 ± 20.17
4	2.73 ± 1.06	1.97 ± 1.21	66.53 ± 11.40	67.48 ± 17.05	30.77 ± 13.69	34.63 ± 16.66	42.48 ± 9.23	45.47 ± 18.77
12	3.37 ± 1.18	2.10 ± 1.22	59.30 ± 10.74	67.15 ± 15.64	26.82 ± 13.13	33.88 ± 16.32	42.12 ± 7.66	46.00 ± 18.58
24	3.43 ± 1.17	2.15 ± 1.18	60.57 ± 10.74	67.08 ± 15.52	26.60 ± 12.88	33.93 ± 15.01	42.33 ± 6.85	46.62 ± 19.14
F	14.71	2.52	5.99	2.20	8.90	2.07	2.31	1.14
*p*	<0.001 **	0.067	0.046 *	0.076	0.031 *	0.078	0.048 *	0.115

Active rTMS group—AG; Sham rTMS group—SG. * *p* < 0.05, ** *p* < 0.001.

**Table 5 brainsci-11-00765-t005:** Comparison of time and group on log transformed BDNF concentration.

Time Points	AG	SG	*t*	*p*	Time	Group	Time × Group
0	4.03 ± 2.00	4.08 ± 1.57	0.07	0.947	F = 10.70	F = 0.52	F = 1.55
2	4.07 ± 1.59	4.16 ± 1.54	0.17	0.732	*p* < 0.001 *	*p* = 0.471	*p* = 0.350
4	4.18 ± 1.39	4.23 ± 1.51	0.36	0.511			
12	4.51 ± 1.34	4.29 ± 1.48	0.89	0.264			
24	4.66 ± 1.29	4.39 ± 1.47	1.29	0.086			
F	31.84	17.65					
*p*	<0.001 *	<0.001 *					

Active rTMS group—AG; Sham rTMS group—SG. * *p* < 0.001.

**Table 6 brainsci-11-00765-t006:** Correlation between log transformed BDNF concentration and HAMD-17, FEIT, HT, and WCST score at each time point in active rTMS group.

		Time Points (Week)	Log Transformed BDNF			
			2	4	12	24
**HAMD-17**		2	−0.005 (0.970)	-	-	-
		4	-	−0.059 (0.669)	-	-
		12	-	-	−0.471 (0.062)	-
		24	-	-	-	−0.788 (0.035 *)
**FEIT**		2	−0.147 (0.263)	-	-	-
		4	-	−0.004 (0.974)	-	-
		12	-	-	−0.020 (0.877)	-
		24	-	-	-	−0.108 (0.411)
**HT**		2	-	-	-	-
		4	-	−0.121 (0.357)	-	-
		12	-	-	−0.077 (0.561)	-
		24	-	-	-	−0.147 (0.263)
**WCST**	**CC**	2	0.011 (0.933)	-	-	-
		4	-	−0.062 (0.636)	-	-
		12	-	-	−0.129 (0.326)	-
		24	-	-	-	−0.258(0.046 *)
	**Re**	2	−0.100 (0.447)	-	-	-
		4	-	−0.041 (0.753)	-	-
		12	-	-	0.016 (0.901)	-
		24	-	-	-	0.089 (0.499)
	**Rpe**	2	0.166 (0.204)	-	-	-
		4	-	0.098 (0.455)	-	-
		12	-	-	0.154 (0.240)	-
		24	-	-	-	0.255(0.085)
	**nRpe**	2	0.122 (0.354)	-	-	-
		4	-	0.065 (0.621)	-	-
		12	-	-	0.183 (0.162)	-
		24	-	-	-	0.244 (0.061)

The contents in the table are correlation coefficient (*p* value). * *p* < 0.05.

## Data Availability

The raw data supporting the conclusions of this article will be made available by the corresponding author.

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
