# Peer review of "Impact of Repetitive Transcranial Magnetic Stimulation (rTMS) on Theory of Mind and Executive Function in Major Depressive Disorder and Its Correlation with Brain-Derived Neurotrophic Factor (BDNF): A Randomized, Double-Blind, Sham-Controlled Trial"

_brainsci, 2021, doi:10.3390/brainsci11060765_

Round 1

Reviewer 1 Report

The question of the correlation of the neurotrophins level in the blood plasma of patients and its level in competent regions of the brain in various diseases of the nervous system remains open. Is there any evidence that rTMS leads to an increase in BDNF level in brain tissues such as hippocamp, amygdala or prefrontal cortex? Could animal data be available?

If  I correctly understand from the text of the article that patients received escitalopram. According to literary data escitalopram has the ability to increase the level of BDNF in blood plasma.  It is possible that the absence of significant correlation with the serum BDNF concentration in the studied  patient groups may be related to this particular fact?

I think that the above issues should be discussed in the text of the article.

Author Response

We are very pleased to receive the comments of the reviewer. Thank you very much for the affirmation of our study.

According to the comments, make the following changes to the manuscript:

1. Is there any evidence that rTMS leads to an increase in BDNF level in brain tissues such as hippocamp, amygdala or prefrontal cortex? Could animal data be available?

  • The related research on the changes of BDNF level in brain tissues of murine models of depression such as hippocampus and prefrontal cortex induced by rTMS has been added to the discussion.

2. According to literary data escitalopram has the ability to increase the level of BDNF in blood plasma. It is possible that the absence of significant correlation with the serum BDNF concentration in the studied patient groups may be related to this particular fact?

  • The level of BDNF increased in both groups, but there is not significant correlatioin with serum BDNF. It may be related to the use of citalopram in both groups. has been added to the discussion.

3. English language and style are fine/minor spell check required.

  • The language quality of the manuscript has been revised again by native English speaker in the same field.

Reviewer 2 Report

Pathophysiology of the major depressive disorder (MDD) particularly in the context of possible hypofrontality is still an important topic in the contemporary neuropsychiatry. Pharmacotherapy of this serious disease with the classical and novel antidepressants did not meet all clinical expectations. On the other hand, some alternative therapeutic strategies such as repetitive transcranial magnetic stimulation (rTMS) may have opened up  several  intriguing fields both in contemporary brain research and applied psychiatry.

This  interesting  and valuable research article by Tong and colleagues reports some benefits of rTMS for patients suffering from MDD,  however no significant correlation with serum BDNF level was found. The main strength of the paper in the first analysis of potenetial correlations between rTMS efficacy and serum BDNF level in patients suffered from MDD.

Despite some llimitations mentioned by Authors, experimental paradigm is in general novel and well considered and all  measurements including Hamilton Depressing Rating Scale, Facial Emotion Identification Test (FEIT) and Hinting Task (HT) were kept the high standard.  Appropriate statistical methods were applied with sufficient number of experimental data. The study is perfectly documented and manuscript is distinctly informative. All tables are also well designed and clear. To sum up, this article may be considered as a valuable contribution to the field of neuropsychiatry and neuroendocrinology.

I have got  just one suggestion for the Authors:

Discussion would benefit from providing a piece of information about possible biochemical mechanism of rTMS therapeutic effect in the treatment of neuropsychiatric disorders.

Author Response

We are very pleased to receive the comments of the reviewer. Thank you very much for the affirmation of our study.

 According to the comments, make the following changes to the manuscript:

1. Discussion would benefit from providing a piece of information about possible biochemical mechanism of rTMS therapeutic effect in the treatment of neuropsychiatric disorders.

  • The biochemical mechanism of rTMS in the treatment of neuropsychiatric disorders has been added to the introduction.

2.  English language and style are fine/minor spell check required.

  • The language quality of the manuscript has been revised again by native English speaker in the same field.